# A mechanistic model of snakebite as a zoonosis: Envenoming incidence is driven by snake ecology, socioeconomics and its impacts on snakes

**Gerardo Martín**[1,2]*, **Joseph J. Erinjery**[3,4], **Dileepa Ediriweera**[5], **H. Janaka de Silva**[5], **David G. Lalloo**[6], **Takuya Iwamura**[7], **Kris A. Murray**[1,8]

**1** MRC Centre for Global Infectious Disease Analysis, Imperial College London, London, United Kingdom, **2** Departamento de Sistemas y Procesos Naturales, Escuela Nacional de Estudios Superiores unidad Mérida, Universidad Nacional Autónoma de México, Mérida, México, **3** School of Zoology, Department of Life Sciences, Tel Aviv University, Tel Aviv, Israel, **4** Department of Zoology, Kannur University, Kannur, India, **5** Faculty of Medicine, University of Kelaniya, Ragama, Sri Lanka, **6** Liverpool School of Tropical Medicine, Liverpool, United Kingdom, **7** Department of Forest Ecosystems and Society, College of Forestry, Oregon State University, Corvallis, Oregon, United States of America, **8** MRC Unit The Gambia at London School of Hygiene and Tropical Medicine, Atlantic Boulevard, Fajara, The Gambia

* gerardo.mmc@enesmerida.unam.mx

**Data Availability Statement:** All data and code are available in the Zenodo repository https://doi.org/10.5281/zenodo.5911958.

## Abstract

Snakebite is the only WHO-listed, not infectious neglected tropical disease (NTD), although its eco-epidemiology is similar to that of zoonotic infections: envenoming occurs after a vertebrate host contacts a human. Accordingly, snakebite risk represents the interaction between snake and human factors, but their quantification has been limited by data availability. Models of infectious disease transmission are instrumental for the mitigation of NTDs and zoonoses. Here, we represented snake-human interactions with disease transmission models to approximate geospatial estimates of snakebite incidence in Sri Lanka, a global hotspot. Snakebites and envenomings are described by the product of snake and human abundance, mirroring directly transmitted zoonoses. We found that human-snake contact rates vary according to land cover (surrogate of occupation and socioeconomic status), the impacts of humans and climate on snake abundance, and by snake species. Our findings show that modelling snakebite as zoonosis provides a mechanistic eco-epidemiological basis to understand snakebites, and the possible implications of global environmental and demographic change for the burden of snakebite.

## Author summary

Snakebite envenoming occurs after contact between two vertebrates, which makes it similar to some transmissible diseases. Based on such similarity, we used estimates of snakebite incidence, snake abundance and biology, and surrogates of human occupational risks to derive a mathematical expression that represents snakebite envenoming as human-snake contacts. Our model explained risk variability very well. We found that snake and human

**Funding:** This research was funded by the Medical Research Council grant number MP/P024513/1, obtained by KM, TI, DGL, HJdeS, and Peter J. Diggle. The funders had no role in study design, data collection and analysis, decision to publish, or preparation of the manuscript.

**Competing interests:** The authors have declared that no competing interests exist.

abundance explain incidence estimates and that agriculture-linked occupations tend to have more frequent contacts with snakes; and that snake abundance decreases with increasing human population. Because snake abundance estimates and contact rates are based on climate or land cover, we identify a pathway for land use change, global warming and population growth to affect the epidemiology of snakebites.

## Introduction

Snakebite envenoming causes acute life-threatening disease with long lasting consequences for survivors [1]. By most recent estimates, up to 1.8 million people suffer from snakebite envenoming every year, of which 20,000–94, 000 die of the resulting illness [2]. Such a high burden is increasingly recognised as a global health crisis and, in combination with its relative neglect from a research perspective, has led to snakebite's recent inclusion on the WHO list of class A neglected tropical diseases (NTDs) [3], a first for a non-infectious disease, and to the development of a global action plan to reduce its burden [4].

Mathematical modelling has been useful for identifying processes that affect the incidence of NTDs and managing them to reduce their burden [5,6]. For instance, modelling revealed that only treating confirmed cases of lymphatic filariasis while neglecting vector and alternative host populations facilitated low-level endemic persistence and the evolution of drug resistant parasites, which together hampered long-term mitigation (reviewed in [7]). Models have also been instrumental for testing and implementing interventions for rabies [8]. The spillover of rabies from its zoonotic mammalian host, mostly as a result of a bite from an infected canine, is a relatively simple transmission process that can be represented with epidemiological models.

Rabies, the above example, and snakebite envenoming have some striking similarities. That is, a pathogenic *agent*—venom—is transmitted to humans after a contact event with a vertebrate host. Despite this similarity, mathematical models of snakebite are scarce (but see [9]), limiting the extent to which mitigation strategies can be assessed prior to field implementation, for example. In contrast, pharmaceutical solutions still dominate mainstream snakebite mitigation and research agendas (e. g., Snakebites: making treatments safe, effective and accessible | Wellcome; [10]). The snakebite roadmap aims to reduce the number of snakebite deaths and disabilities by 50% by 2030 [11], and identifies novel methods and tools to better understand snakebite epidemiology as a priority to help achieve this goal. Improved epidemiological models for snakebite could fill a major current void in understanding snakebite, improving mitigation efforts and maximising the efficacy of post-bite treatment systems (e.g., directing antivenom supplies efficiently) [5,12].

Mechanistic representation of zoonotic spillover predicates that transmission depends on three major principles [13,14]: 1) reservoir and spillover hosts coinciding in time and space (e.g. [15]), 2) the disease prevalence and/or pathogen shedding in reservoir hosts depending on the contact route necessary (e.g. [16]), and 3) the spillover host developing disease when it is infected. Risk mapping studies of snakebites indicate that there are analogous factors relating to these mechanisms for zoonotic spillover. First, models of snake distribution and abundance, and maps of human population density can be used to represent human-snake spatial [17,18] and temporal [19] alignment, which underlies human-snake contact. Second, the equivalent of pathogen prevalence (i.e,. possessing venom) is 100% among venomous snakes, hence the likelihood of transmission (envenoming) is rather a function of how likely it is that venom is delivered during a bite (as opposed to dry bites), which varies considerably between venomous species [20]. And third, host susceptibility relates to the effect of venoms on the human body [21].

Exposure factors and post exposure vulnerabilities (e.g. occupational, cultural and socioeconomic) may mediate these steps to further influence outcomes and ultimately the overall burden in a population [22–24]. For instance, working and living in a agrarian setting could increase spatio-temporal alignment; poverty could limit the use of protective clothing and also exposure to snakebites due to fragile dwellings made of locally found materials, that can increase the probability of envenoming inside houses given a bite; and cultural differences could influence healthcare seeking behaviour, which could determine the toll on the body during an envenoming event. Fig 1, following Plowright et al.,'s [13] framework for zoonotic spill-over, represents such a conceptual arrangement as a sieve that starts with snake diversity, distribution, abundance, behaviour, venom toxicity, and its overlap with humans and individual risk/susceptibility factors.

Whereas the majority of NTDs comprise transmission of a pathogenic microorganism (or a complex of microorganism species e.g., *Leishmania spp*.) from one or more reservoir and/or vector hosts to humans, the transmission dynamics of snakebite involve envenoming of humans (or other victim, such as livestock) from one or more venomous snake ('reservoir') species. In systems in which only one species dominates the snakebite burden, an appropriately simple single-species model has been shown to successfully predict the geographical

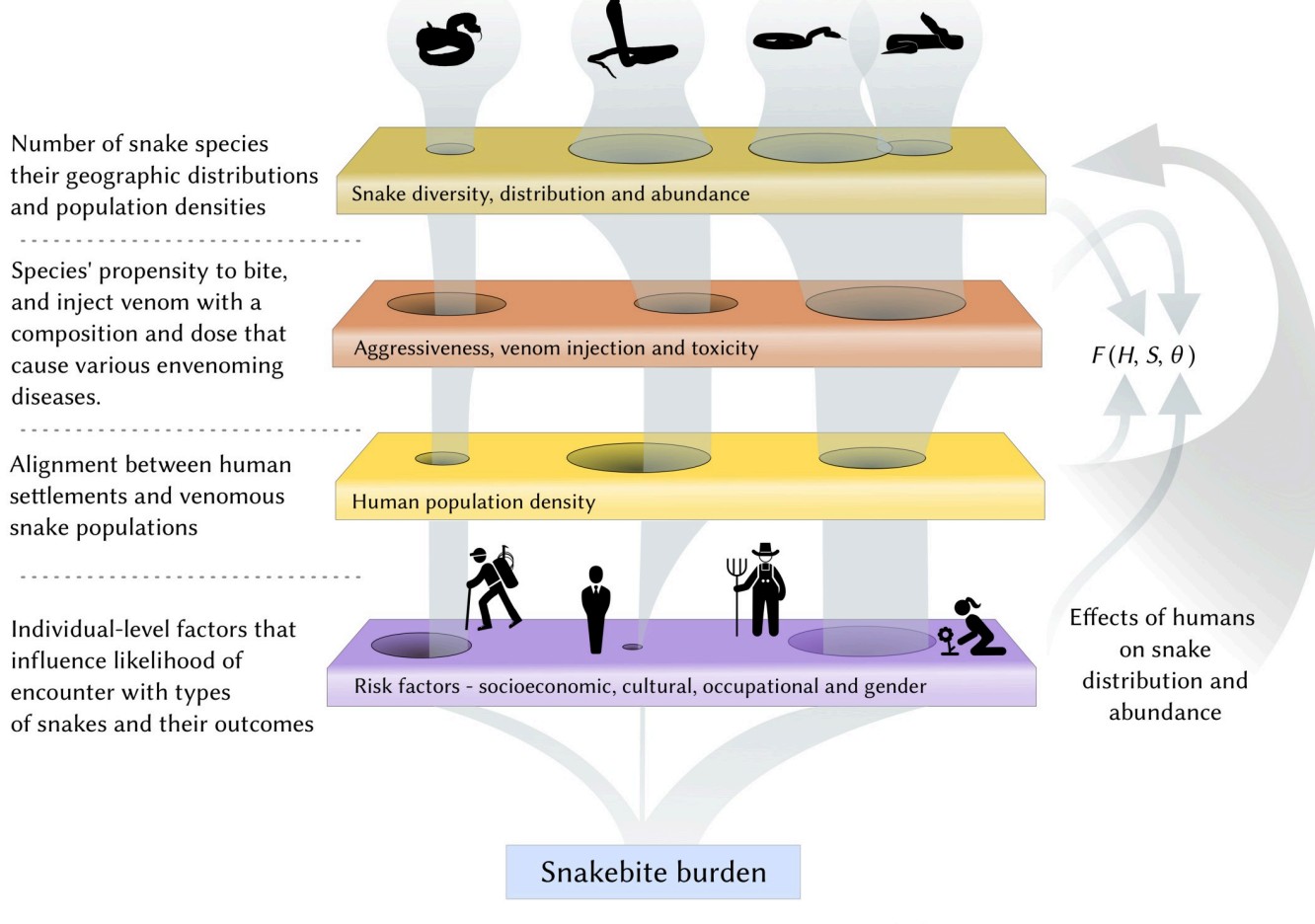

**Fig 1. The snakebite sieve, an adaptation of the conceptual framework for zoonoses of Plowright et al. [13], shows how different factors align and result in what we recognise as snakebite burden.** Species and occupational identities are meant to be schematic and do not represent the characteristics of our study system.

variability of snakebite incidence. For instance, Bravo-Vega et al. [9] modelled the frequency of encounter with *Bothrops asper* as a function of environmental suitability to predict snakebite incidence in Costa Rica. However, such models have not been applied in settings where multiple species bite and influence variability of snakebite envenoming incidence in space and in time. For example, the 'big four' species considerably shape the burden of snakebite envenoming in South Asia [25], while Goldstein et al. [26] show that burden dynamics can be complex over time due to the influence of multiple biting species and the environment on the behaviour of both humans and snakes.

In the present study, we sought to capitalise on recent snakebite research developments in incidence mapping [23], snake distributional ecology [18], and zoonotic spillover theory [13] to develop a novel mechanistic epidemiological framework representing the biological components of snakebite. Specifically, we explored the extent to which various types of models typically applied to directly transmitted infectious diseases explain the geography of snakebite and envenoming incidence estimates. We use the island nation of Sri Lanka, a snakebite hotspot, as a model system given that, as of the time of writing, it is the only high snakebite burden region for which high quality/high resolution country-wide data exist to formulate, test and compare models on snakebite and envenoming incidences. In addition to demonstrating successful risk mapping, we show that geographical patterns of incidence arise from dynamic environmental socio-ecological processes that include effects of climate, occupational risk factors, land use and its changes and direct human impacts on snake populations. Recasting snakebite as a zoonosis and formally applying conventional epidemiological models provides a novel way forward in understanding snakebite epidemiology, which in future studies would allow us to better anticipate risks and potentially help achieve ambitious mitigation targets [11] in a rapidly changing world.

## Methods

We first identified a series of mathematical formulations for human-snake contacts to represent snakebite as a zoonotic disease transmission process. In conventional infectious disease transmission models, disease spread is considered to be frequency-, density-dependent or a mixture of both. When frequency-dependent, the per-capita rate at which susceptible individuals become infected depends on pathogen prevalence in the population. When density-dependent, transmission increases with infected host density. For snakebites, pathogen prevalence for a given reservoir is 100% (all individuals of a venomous species carry venom; Fig 1). Frequency dependence might function for snakebites in the presence of dry bites, such as occurs when venom glands are depleted [20]. However, its causes are still poorly understood to be incorporated in a process-based model. The remaining density-dependent contact formulations (Table 1) summarise different mixing dynamics between humans and snakes, resulting in functional relationships between snake abundance and snakebite incidence that range from linear to asymptotic to bell-shaped [27]. To test the models' abilities to explain snakebites, we transformed them from continuous to discrete time, representing annual trends without seasonality, and estimated their parameters regressing the estimated snakebite and envenoming incidence rates from Ediriweera et al. [23] against each functional relationship. As Ediriweera et al. [23] estimates are spatially explicit we first fitted a model ignoring the spatial component, and then we fitted the same models with a conditional autoregressive random effect.

### Data

Data used for fitting and testing models and estimating parameters were two published datasets (rasters) of the spatial distribution of snakebite and envenoming incidence rates, estimated

**Table 1. Disease transmission terms tested representing functional relationships between snakes and humans and resulting in snakebite.** Each term has been previously applied in various settings for infectious disease studies so here we list only the earliest reference for each. DIC = deviance information criterion, pD = potential degrees of freedom, $H$ = humans, $S$ = snakes, $\beta$ = contact rate. Missing DIC values means that we could not obtain a converged model, so DIC values were not comparable with those of converged models.

| Tranmission term | Discrete time form $F(H, S, \theta)$ | Name and description | Source | DIC, pD* |
|---|---|---|---|---|
| $\beta HS$ | $1 - \exp(-\beta S)$ | **Simple mass action**. $H \times S$ is the total number of possible contacts | [30] | 17484, 20.02 |
| $\beta H^p S^q$ | $1 - \exp(-\beta\, H^{p-1} S^q)$ | **Power**. Similar to mass action, but $p$ and $q$ take values 0, 1, and increase or decrease the number of contacts of S in relation to H and vice-versa. | [42] | - - - |
| $\beta H (S - H/q_h)$ | $1 - \exp(-\beta\, (S/H - 1/q_h))$ | **Refuge effect on S**. Parameter $q$ represents the fraction of snakes exposed to humans depending on the number of humans present | [43] | - - - |
| $\beta S (H - S/q_s)$ | $1 - \exp(-\beta S\, (1 - S/(q_s \cdot H)))$ | **Refuge effect on H**. Parameter $q$ represents the fraction of humans exposed to snakes depending on the number of snakes present | [43] | 17062, 26.0 |
| $\beta HS \cdot \begin{cases} \dfrac{S}{1 - \varepsilon + \varepsilon S} \\ \dfrac{H}{1 - \varepsilon + \varepsilon H} \end{cases}$ | $1 - \exp\left(-\beta S \cdot \begin{cases} \dfrac{S}{H - \varepsilon H + \varepsilon HS} \\ \dfrac{1}{1 - \varepsilon + \varepsilon H} \end{cases}\right)$ | **Separate asymptotic** term on $H$ or $S$, where $0 \leq \varepsilon \leq 1$. When parameter $\varepsilon = 0$, the expression reduces to the simple mass action model, otherwise the effect of $H$ or $S$ becomes asymptotic. | [44] | - - - |
| $\beta HS \cdot \begin{cases} \dfrac{1}{c + S} \\ \dfrac{1}{c + H} \end{cases}$ | $1 - \exp\left(-\beta S \cdot \begin{cases} \dfrac{1}{cH + SH} \\ \dfrac{1}{cH + H^2} \end{cases}\right)$ | **Asymptotic** on $H$ or $S$. Parameter $c$ represents the number of $S$ or $H$ at which 50% of the contacts occur. | [44] | - - - |

with model-based geostatistics applied to a country-wide community survey of ~0.8% of the Sri Lankan population [23], under the assumption that these estimates represent the ground truth. The response variables for regressing the functional relationships were the number of snakebite and envenoming cases, respectively, found by multiplying the mapped incidence rates by human population density (described below). The independent data used to explain incidence rates and which represent the causal snakebite factors (Fig 1) were:

**Distribution and abundance of reservoir hosts.** Raster images of the abundance patterns of the most medically relevant venomous snakes of Sri Lanka (three elapids and four vipers), estimated with point process models as functions of the environment (climate, topography and land cover), and adjusted for species' relative abundances. Species records to build abundance models span several decades of fieldwork across the island [28].

**Distribution and abundance of spillover hosts.** Raster image of human population density raster layer from 2010 (closest point in time prior to the snakebite survey period of August 2012 –June 2013; [23]) obtained from the Gridded population of the world (GPW v4) hosted by the Socioeconomic Data and Applications Center (SEDAC, https://sedac.ciesin.columbia.edu).

**Spillover host exposure risk factors.** Raster image of land cover representing the predominant classes forest, degraded forest, agriculture, urban and tea (see 'Deriving land cover data' below for source details). Land cover correlates well with socioeconomic status and predominant occupation [29], which are the primary human-related risk factors [22] and which in the model represent different risk categories via model parameters such as the human-snake contact rate.

## Data formatting

Prior to analysis we homogenised and synchronised the resolution of all data to a common grid comprising $5 \times 5$ km pixels ($25$ km$^2$) projected to the datum of Sri Lanka (SLD99, EPSG 5235). Synchronising data allowed matching data points to regress the number of snakebite and envenoming cases against human, snake and land cover data according to the model

tested. A 5 × 5 km grid was chosen to facilitate computation and retain a reasonable degree of biological detail relevant to the study aims. Human population density and snake abundance estimates were upscaled from their original 1 km resolution by aggregation, summing the values of adjacent cells by a factor of 5 grid cells along longitude and latitude. Snakebite and envenoming incidence layers were resampled from their original ~1.5 × 3 km to the target resolution using weighted bilinear interpolation. The land cover layer was upscaled from 30 m to 5 km by majority vote per pixel (land cover class was assigned to each grid cell based on the most common class among the ~27000 30 m grid cells contained in each 5 × 5 km pixel).

### Deriving land cover data

The five categories considered for the analyses were derived using unsupervised isoclustering and visual interpretation of remotely sensed data for the year 2010 (Landsat surface reflectance, original Landsat optical bands and NDVI). The resulting 30 m land cover maps were validated using 600 randomly generated points across Sri Lanka, with which we estimated a classification accuracy of >95%.

### Functional relationships for human-snake contacts

The simplest formulation of human-snake contact is the mass action model, whereby the total number of possible different contacts is found by *No. Humans × No. Snakes*. This formulation is widely used for zoonotic transmission, and the simplest model relevant to snakebites is the *susceptible-bitten* model [30]. Here, the growth of bitten humans ($H_b$) per time unit ($dH_b/dt$) is proportional to the number of possible contacts between susceptible humans ($H_s$) snakes ($S$):

$$\frac{dH_s}{dt} = -\beta H_s S \tag{1.1}$$

$$\frac{dH_b}{dt} = \beta H_s S \tag{1.2}$$

where $\beta$ is the human-snake contact rate. This model assumes that time is continuous, however snakebite incidence data has a resolution of one year, for which discrete-time models are more adequate. The continuous (left) and discrete-time (right) models for snakebite based on the SB model are:

$$\frac{dH_s}{dt} = -\beta H_s S \Rightarrow H_{t+1} = H_{s,t} - H_{s,t}\exp(-\beta S) \tag{1.3}$$

$$\frac{dH_b}{dt} = \beta H_s S \Rightarrow H_{b,t+1} = H_{b,t} + H_{s,t}(1 - \exp(-\beta S)) \tag{1.4}$$

The tested human-snake contact formulations are given in Table 1 in their original and discretised forms. We held no *a priori* reason to favour one model over another, so we explored all of their relative abilities to explain snakebite and envenoming incidence.

### Model implementation and selection

Ediriweera et al., [23] report both snakebite and snakebite envenoming incidence nationally for Sri Lanka, which here we refer to as two measures of risk, Snakebite and Envenoming. The former represents all confirmed contacts with snakes with and without envenoming, and the latter are the contacts which resulted in envenoming illness. To model envenoming we used two approaches. First, we modelled snakebite as a contact process with the functional

relationships and assumed that envenoming is a subset or secondary event, whose probability of occurring was another function of snake abundance. Second, we treated envenoming cases alone in the same way we treated snakebites (see below).

To estimate model parameters we regressed the number of snakebite or envenoming cases against the functional relationships (Table 1) using Markov Chain Monte Carlo (MCMC) sampling in JAGS [31] via R2Jags package in R [32]. Prior to implementation, the contact processes (Table 1, column 1) were transformed from their continuous-time form into discrete-time, representing the probability that there were any snakebites during the study period ($t$, $t$ +1). The discrete-time form of Eq 1.4 to represent snakebite incidence is [33]:

$$H_{b,\,t+1} - H_{b,\,t} = H_{s,\,t} \times (1 - \exp(-\beta S_t)) \tag{2.1}$$

Therefore, the probability that there are any snakebites when during one year ($t$, $t$ +1) is:

$$P(t, t+1) = \frac{H_{b,t+1} - H_{b,t}}{H_s} = 1 - \exp(-\beta S)$$

To summarise, after taking $H_s$ from the right to the left hand side of the equations, snakebite or envenoming incidence is proportional to a function of snakes and/or humans, $F(H, S) = P(t, t+1)$. More generally the number of snake-bitten people is:

$$H_b = H_s \times P(t, t+1)$$

To select a more appropriate model in the MCMC sampling process we treated snakebite and envenoming cases as either Poisson:

$$H_b* \sim \text{Poisson}(H_b)$$

or Negative Binomial, parameterised by $P_N$ and dispersion parameter $r$:

$$r \sim \text{unif}(0, 50)$$

$$P_N = \frac{r}{r + H_b}$$

$$H_b* \sim \text{NegBin}(P_N, r)$$

in order to use the best statistical distribution for the number of cases.

## Human-snake contact rates

To estimate effects of the different snake species and human-related factors, the contact rate β was decomposed into different aspects. The first aspect included the estimation of one contact rate per snake species, such that $S_t$ from Eq 2.1, for instance is:

$$S_t = \sum B_s \times (A, E)_s \times S_s$$

the sum of the individual snake species abundances multiplied by their specific human-snake contact rates $B_s$, and species aggressiveness or envenoming-severity indices (($A$, E)$_s$; [28]). Each index was included depending on whether the contact process analysed was snakebite ($A_s$) or envenoming ($A_s \times E_s$). This means that the absolute magnitude of contact rate for snakebite is $B_s \times A_s$ and for envenoming is $B_s \times A_s \times E_s$. Thus $B_s$ represents other factors not related to aggressive behaviour ($A_s$) or envenoming severity ($E_s$) that influence human-snake interactions and their outcomes.

The second component of the decomposed contact rates are a series of functions of human population density (number of people per grid cell) that attempt to adjust total snake abundance in relation to humans, since high human population density is a well known threatening process for biodiversity [34,35]. The functions of human population density were:

$$\beta(H) = \begin{cases} \exp(\beta_0 + \beta_1 H) \\ \exp(\beta_0 + \beta_1 H^2) \\ \exp(\beta_0 + \beta_1 H + \beta_2 H^2) \end{cases} \tag{2.2}$$

In this approach, if $\beta_{0,1,2}$ coefficients are drawn from a normal distribution in the MCMC sampling process it is possible to model the negative effect of humans on incidence, whilst retaining a positive relationship between the total number of cases and human population density since $\beta(H)$ will always be positive. To estimate human susceptibility in the models we categorised their parameters ($\beta$ and 2.2) in five land cover classes.

To summarise, we estimated parameters and measured the ability to represent snakebite and envenoming of the models with and without the functions of human population density (2.2), with and without its parameters categorised by land cover, and estimating a global $\beta$ with and without categorisation by land cover.

## Model of envenoming probability

For the approach of treating envenoming as an event that follows a snakebite, we treated the number of envenoming cases as the subset of snakebites that result in envenoming. Therefore, the number of envenoming cases is:

$$H_{envenomed} = H_b \times P_{env}$$

where $H_{envenomed}$ is the number of envenoming cases and $P_{env}$ is the probability that a snakebite results in envenoming, which we derived from an expert-collated index of species' envenoming severity (Table 2; [28]). To estimate $P_{env}$ we treated the number of envenoming cases as Poisson or Negative binomial, but estimated the probability logistically using two model-

**Table 2. Parameter estimates for the mass action models for snakebites (median and standard deviation).** $\beta_{i,l}$ parameters are those of the function of human population density to adjust total snake abundance in relation to humans in each land cover class. $r_l$ are the negative binomial dispersion parameter for each land cover class. $B_s$ are the estimated contact rates for each snake species after adjusting the point intensities for relative abundances and weighting for aggressiveness behaviour. The star * indicates the parameters whose 95% credible intervals do not contain zero (only relevant to $\beta_{i,l}$). Rows with grey background show estimates for the spatial version of the model. *H. spp* corresponds to *Hypnale spp*, and *T. trig.* to *T. trigonocephalus*. $\theta_{LC}$ denotes parameters related to land cover and $\theta_{Spp}$ denotes parameters for snake species.

| $\theta_{LC}$ | Agriculture | Degraded | Forest | Tea | Urban | | |
|---|---|---|---|---|---|---|---|
| $\beta_{0,l}$ | -12.9 (0.19)* | -12.84 (0.2)* | -13.18 (0.21)* | -11.91 (0.26)* | -11.27 (0.61)* | | |
| | -12.9 (0.11)* | -12.87 (0.11)* | -13.11 (0.12)* | -11.84 (0.15)* | -11.32 (0.2)* | | |
| $\beta_{1,l}$ | -0.004 (0.001)* | -0.005 (0.001)* | -0.000 (0.002) | -0.017 (0.002)* | -0.028 (0.002)* | | |
| | -0.004 (0.001)* | -0.006 (0.001)* | -0.001 (0.001) | -0.018 (0.001)* | -0.027 (0.002)* | | |
| $r_l$ | 23,70 (2.95) | 16.82 (1,13) | 9.32 (1,16) | 7,18 (0.69) | 12,10 (4.52) | | |
| | 34.47 (2.33) | 23.56 (0.91) | 13.25 (0.93) | 9.39 (0.56) | 18.85 (7.02) | | |
| $\theta_{Spp}$ | *B. caeruleus* | *B. ceylonicus* | *D. russelli* | *E. carinatus* | *H. spp* | *N. naja* | *T. trig.* |
| $B_S$ | 3.48 (0.56) | 5.30 (2.85) | 1.17 (0.37) | 2.99 (1.07) | 0.036 (0.02) | 8.48 (1.29) | 0.17 (0.16) |
| | 4.34 (0.48) | 5.58 (2.26) | 1.72 (0.39) | 2.81 (0.97) | 0.047 (0.015) | 3.51 (0.69) | 0.14 (0.11) |

formulas:

$$\log\left(\frac{P_{env}}{1 - P_{env}}\right) = \sum B_s \times E_s \times S'_s \tag{3.1}$$

$$\log\left(\frac{P_{env}}{1 - P_{env}}\right) = B_{LC} + \sum B_s \times E_s \times S'_s \tag{3.2}$$

where, in both equations, $B_s$ is the statistical effect of snake species $s$, $E_s$ is the index of envenoming severity (Table 2) and $S'_s$ is snake species $s$ after applying the correction of abundance in relation to humans ($\beta(H)$). In model formula 3.2, the term $B_{LC}$ is a random intercept for land cover class. The criteria to select one formula over the other was minimisation of the deviance information criterion (DIC) and convergence [36].

## Model selection

We first implemented all functional relationship models and ran short MCMC chains of 10–50 K iterations to discard those with erratic sampling behaviour, poor chain mixing or consistently higher DIC than other functions. Once we obtained a more manageable subset of models we ran the full MCMC chains of up to 750,000 iterations and then checked for convergence with the Gelman diagnostic test [37]. With the latter we made sure that posteriors are unrelated to starting prior values (ratio of between and within chain variances should approach one). Finally, we analysed the spatial pattern and statistical distribution of the residuals by subtracting the median of posterior samples with snakebite and envenoming incidence data.

The criteria to select one functional relationship over another were: 1) ease of parameter estimation and convergence of MCMC chains; 2) adequate reproduction of the spatial pattern of raw number of snakebites and envenomings and their annual incidence rates, by measuring the spatial association of the predictions with a modified T-test for spatially autocorrelated data [38]; 3) minimising the DIC; and 4) adequate representation of the statistical distribution of the snakebite and envenoming data using quantile-quantile plots.

## Model with spatial effects

To fit the models with spatial effects we used the mean and standard deviation estimates of the non-spatial version as parameter priors for the main effects $\beta_0$, $\beta_1$, and $B_S$ to ease convergence. Random effects for the above models were incorporated as a log-linear random intercept for the number of predicted cases:

$$\log H_b = \log(H \times F(H, S)) + \rho_i$$

This means that $\rho$ was sampled from a conditional autoregressive normal distribution where each $\rho_i$ is proportional to the average $\rho_{-i}$ of its immediate neighbours in a queen-type neighbourhood. These final models were fitted with Nimble [39] instead of JAGS. Geographical data were manipulated with the raster and rgdal R packages [40,41].

## Results

The selected models reproduced well the magnitude and distribution of both snakebite and envenoming incidence rates observed in Ediriweera et al., [23]. National snakebite and envenoming incidence patterns were best predicted by first modelling snakebites with the functional relationship based on simple mass action (Table 1) and then estimating the probability that a snakebite results in envenoming using the land cover random-intercept model (Eq 3.2).

## Snakebites model

Of the contact formulations listed in Table 1, the simple mass action ($\beta SH$) and refuge effect on humans ($\beta S(H-S/q)$) were the best performing models, with the lowest deviance information criterion (DIC; Table 1). The refuge effect model had lower DIC than the simple mass action (Table 1) and higher correlation with the snakebites and incidence data (refuge effect, $r = 0.67$, d.f. = 128, $P = 0$; simple mass action, $r = 0.61$; d.f. = 137; $P = 0$), suggesting at face value a better fit. We considered that the remaining formulated models were all unsuitable as we failed to obtain reliable parameter estimates due to lack of MCMC convergence.

To select one model over another as the better one, we also took into account the statistical distribution of predicted incidence rates. The statistical distribution of the incidence rates produced by the refuge effect model was very different to the original incidence rate distribution (S1 Fig). Therefore, we chose the simple mass action model. The number of snakebites and incidence patterns produced by this model were qualitatively very similar and had a statistical distribution nearly identical to that of the data (S1 and S2 Figs).

The spatial version of the simple mass action model converged with 1M iterations, and had an even higher correlation with Ediriweera et al. [23]'s estimates, $r = 0.87$ (d. f. = 81, $P = 0$) for incidence and $r = 0.97$ (d. f. = 29, $P = 0$) for the number of bites (Fig 2).

The decomposition of contact rates that worked best was estimating a contact rate for each snake species and correcting snake abundance in relation to human population density by land cover class and log-transforming human population density:

$$\beta(H, L) = \exp(\beta_{0(l)} + \beta_{1(l)} \ln(H)^2) \tag{4.1}$$

All the parameters of the spatial and non-spatial simple mass action model converged (Table 2) and did not exceed the very strict threshold of 1.05 of the Gelman test (very similar variances between and within chains; S1 Table). The estimated effects of humans on snake abundance resulted in different responses of incidence rates to human population and snake abundance in each land cover class (Fig 3).

## Model parameters

Parameters of the correction of snake abundance in relation to human population density ($\beta_{0(l)}$ and $\beta_{1(l)}$, Eq 4.1) were all significantly negative (Table 2), indicating that humans tend to decrease snake abundance in all land cover classes. Also, given significant differences between land cover classes, the effect of humans on snakes depends on predominant land cover class. The largest effect was estimated for *tea* and *urban* cover (snakes decrease faster with human population density), intermediate in *forest* and *degraded forest*, and smallest in *agriculture* (snakes decrease the least with human population density, Table 2).

Regarding individual species' contact rates, the Indian cobra (*Naja naja*) had the highest estimated rate, followed by the Ceylon krait (*Bungarus ceylonicus*) and the common krait (*B. caeruleus*). The lowest contact rate was estimated for the hump-nosed viper (*Hypnale spp*). These contact rates showed no relationship with species' relative abundances, indicating that factors not represented in the decomposed contact rates (i.e., frequency of venom injection, time of activity aligned with human activities) are critical but unobserved determinants of risk patterns. Note, that the estimated rates are not statements of how frequently humans encounter them. Estimated parameters should only be interpreted as indices of species' importance for snakebites relative to their abundance and that of humans where they occur. Also, these parameters are negatively correlated with those estimated to adjust abundance in relation to humans. Every order of magnitude of increase in the size of the $B_s$ contact rates must be

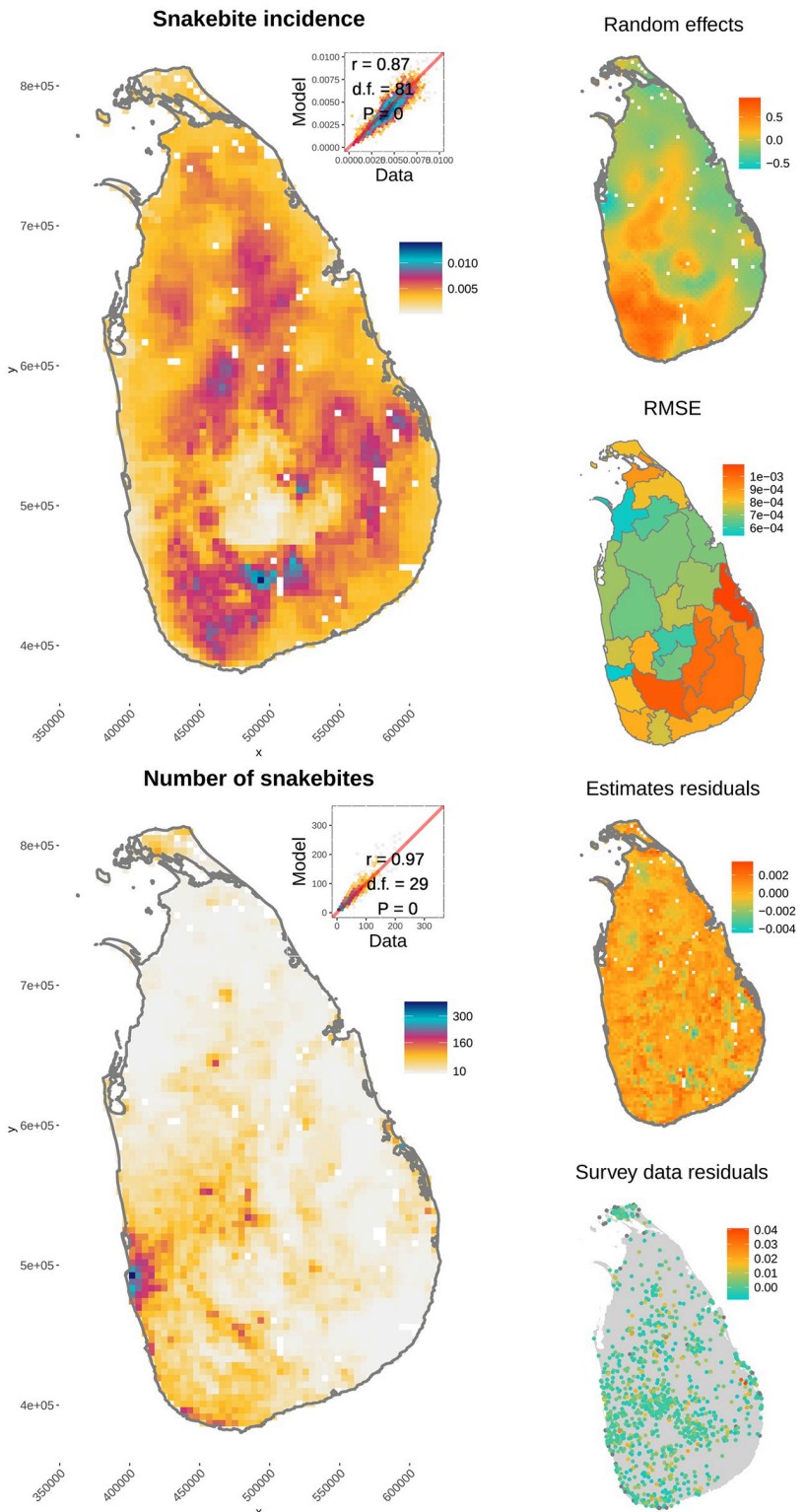

**Fig 2. Snakebite patterns predicted by the spatial model (median of posterior estimates).** Insets in the top right corner of each map show a scatter plot of our model estimates and Ediriweera et al. [23]. Right hand side panels, from top to bottom show the autoregressive random effects, root mean square error for each Sri Lankan district between our model and data used, and the residuals with original survey data analysed by Ediriweera et al. [23].

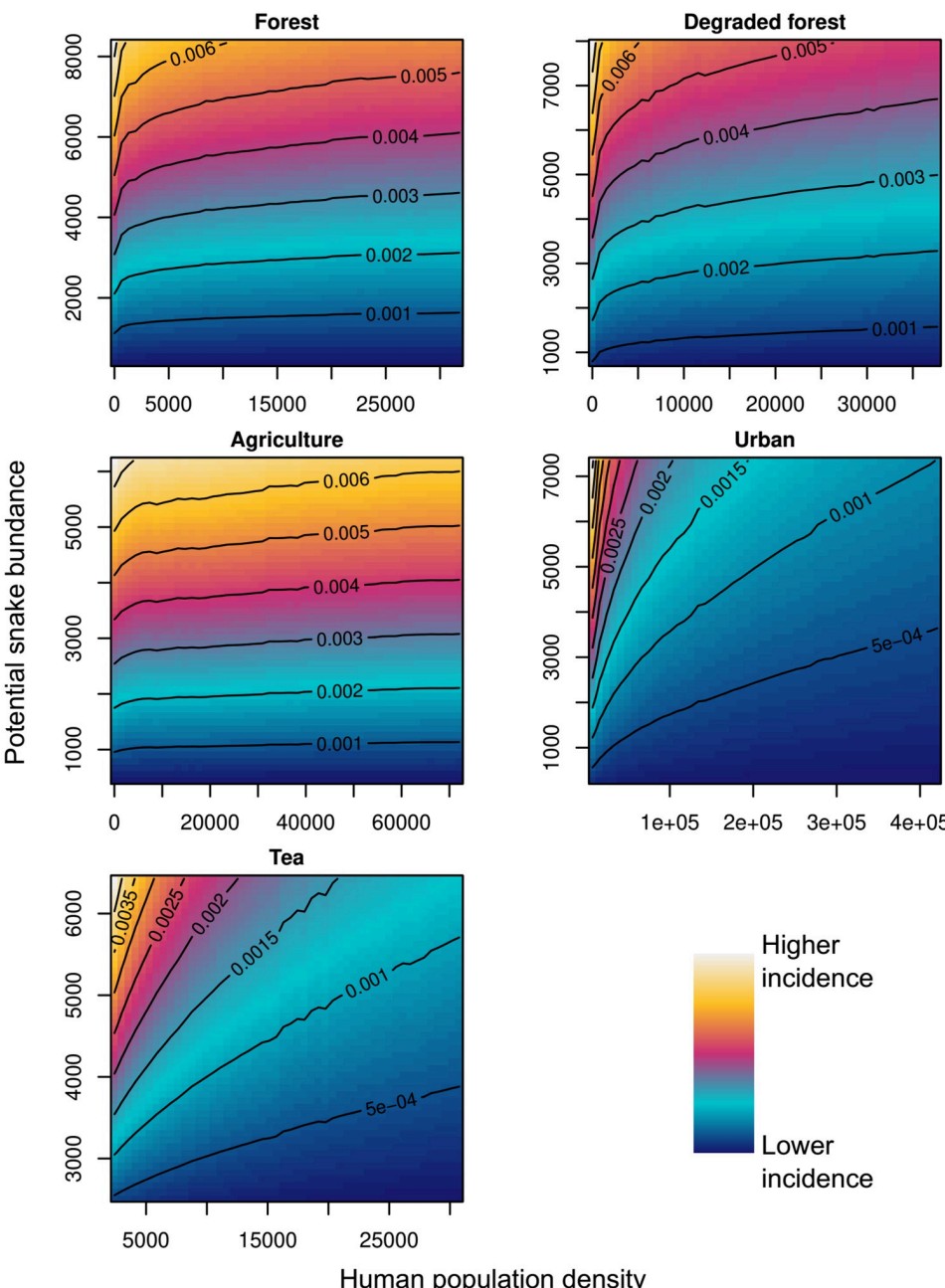

**Fig 3. Partial responses of snakebite incidence to human population density and snake abundance per land cover class in the model of snakebites.** The contour lines in each 2D plot indicate the expected incidence level highlighted with the colour scale at that combination of human population and snake abundance.

accompanied by an absolute decrease in log-scale of the value of $\beta_{0(l)}$ parameter of Eq 4.1 (e.g. $B_s \times 10 \rightarrow \beta 0(l) - \log(10)$, or $B_s \div 10 \rightarrow \beta_{0(l)} + \log(10)$), allowing the interpretations above for the individual species. The magnitude of these parameters indicates the average number of snakes in logarithmic scale that are lost for every human, resulting in faster declines in *urban* land cover, and slowest in *agriculture* (Table 2). The effect of both humans and snakes on snakebite incidence in each land cover class is represented visually in Fig 3.

### Envenoming model

The fact that the random-intercept envenoming model (Eq 3.2) was the more adequate, indicated that land cover influenced the probability that a snakebite resulted in envenoming. The correlation between envenoming incidence predicted by the spatial model and the data was $r = 0.85$ (d. f. = 23, $P = 0$; Fig 4, right panel) and for the number of envenoming cases was $r = 0.93$ (d.f. = 9, $P = 0$; Fig 4, top left panel). Convergence statistics for the spatial model are given in S2 Table, S3 and S4 Figs.

The snake species that had the largest significantly positive fitted effect on the probability of envenoming was *Bungarus caeruleus*, indicating that this species explains most of the spatial heterogeneity of the probability that bites result in envenoming in Sri Lanka (Table 3). *D. russellii* and *N. naja* also had significant effects in the outcome of snakebites but were significantly negative, indicating that risk of envenoming after a bite decreased with their predicted abundances (we address this counter-intuitive result in the Discussion). The remaining species' effects were not significantly different from zero, indicating that their contributions towards envenoming cannot be distinguished from random (Table 3).

Land cover classes also had significant effects on the probability that snakebites resulted in envenoming. *Agriculture*, had the largest significantly positive effect followed by *urban*. D*egraded forest* had the largest significantly negative effect, followed by *tea*. The only land cover class with a non-significant effect was *forest*, which suggested that envenoming after a snakebite is more likely to be a function of the biting snake than of the environmental or social context of snake-bitten people in *forest* environments (Table 3, and see Discussion on the role of land cover).

## Discussion

Mathematical models have been critical tools for understanding and controlling the transmission of zoonotic diseases, but despite many ecological and epidemiological similarities few such models exist for snakebite (e. g. [9]). Here, we mapped snakebite and envenoming incidence using a mathematical model that represents human-snake interactions and their outcomes, adapting a mass action model usually applied to the transmission of infectious diseases [30]. We treated venom as pathogenic agent transmitted between venomous snakes and susceptible humans, and tested various functional relationships describing the human-snake contact process and its outcomes. Incidence rates were successfully mapped by estimating contact rates between all medically relevant snakes of Sri Lanka and humans, and by accounting for human and snake factors known to be important determinants of snakebite and envenoming incidence. Human factors (social, economic and cultural) were included categorising parameters by land cover serving as a socio-economic proxy [29]. Snake factors included biological characteristics of the different species, like aggressiveness and severity of the envenoming illness produced by its bite. Furthermore, parameters in the model are sensitive to climate, land cover and topography via their effects on snake population estimates [28]. As such, we have developed a generalisable epidemiological model for snakebite that could be transferrable (given local data) to future environmental conditions.

Predicting spatial and temporal patterns (including future changes) of snakebite risk is possible with purely statistical methods or by focusing only on one or two components of snakebite risk (e.g, [18,45]). An advantage of our approach, however, is that it is process-based and generalisable, explicitly capturing the relevant snakebite processes (e.g., human-snake contact patterns, biological traits of venomous species) to predict epidemiologically meaningful measures of snakebite risk. Such an approach may be more transferrable (i.e., when forecasting burden in other regions) and better suited to forecasting impacts of global change (e.g.,

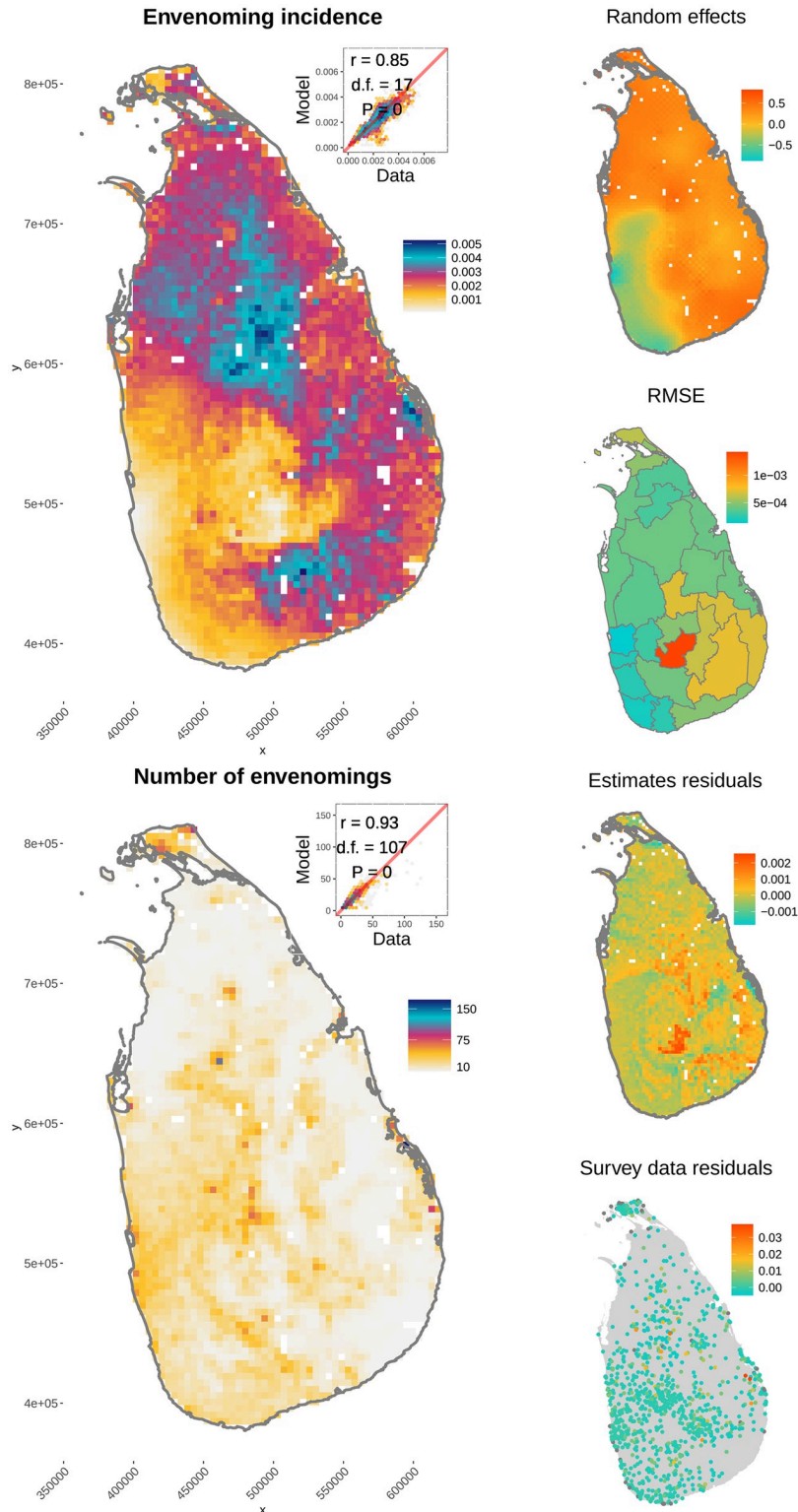

**Fig 4. Envenoming patterns predicted by the spatial model of the probability that a bite results in envenoming (median of posterior estimates).** The top right corner of the left side maps show the relationship with the data used to fit the models, and the correlation coefficient adjusted for spatial autocorrelation. Right hand side panels, from top to bottom show the conditional autoregressive random effects, root mean square error for each Sri Lankan district between our model and data used, and the residuals with the original survey data analysed by Ediriweera et al. [23].

**Table 3. Parameter estimates of the model for probability of envenoming (median and standard deviation).** *Intercept* are the estimated effects of each land cover, and $B_s$ are snake species' effects. The * symbol indicates that the 95% credible intervals of posterior samples do not contain zero. Rows with grey background show estimates for the spatial version of the model. *H. spp* corresponds to *Hypnale spp*, and *T. trig.* to *T. trigonocephalus*. $\theta_{LC}$ denotes parameters related to land cover and $\theta_{Spp}$ denotes parameters for snake species. Negative binomial parameter *r* estimates were 18.59 (1.09) and 49.69 (0.41) for the non-spatial and spatial models respectively.

| $\theta_{LC}$ | Agriculture | Degraded | Forest | Tea | Urban | | |
|---|---|---|---|---|---|---|---|
| Int | 0.255 (0.07)* | -0.394 (0.05)* | 0.056 (0.07) | -0.72 (0.04)* | 0.59 (0.17)* | | |
| | -0.05 (0.04) | -0.33 (0.03) * | -0.23 (0.05)* | -0.48 (0.03) * | 0.53 (0.11)* | | |
| $\theta_{Spp}$ | *B. caeruleus* | *B. ceylonicus* | *D. russelli* | *E. carinatus* | *H. spp* | *N. naja* | *T.trig.* |
| $B_S$ | 280.14 (13.99)* | -15.77 (32.12) | -97.34 (28.4)* | 5.49 (31.91) | -24.546 (23.74) | -290.34 (29.84)* | -2.88 (31.80) |
| | 197.88 (13.41)* | -3.86 (31.68) | -165.129 (25.61)* | 5.22 (31.98) | -34.24 (19.77) | -373.43 (27.66) * | -1.64 (31.59) |

including climate change, land-use change and socio-economic development), where a number of complex and potentially interactive mechanisms could push burden in different directions. However, the success of such applications still depends on the availability of relevant data to parameterise the model. The raw material to build our model were snake occurrence records and behavioural trait data [28]. Improving and applying similar frameworks in other regions therefore requires reliable occurrence data in relation to human settlements and improved ecological information on venomous snakes [28,46]., both of which can be regarded as high priorities for future work [47].

To the best of our knowledge, the only previous study of a mathematical model for snakebite comparable to ours, is Bravo-Vega et al. [9]. We considerably extend their approach by decomposing contact rates to incorporate both human and snake factors. The first snake aspect of the decomposed contact rate $\beta$ represents known (aggressiveness–$A_s$–and envenoming–$E_s$–indices) and unknown biological aspects (estimated statistically) of multiple species. The unknown estimated snake factors are likely related to the alignment of human-snake activity periods (Table 2 for snakebite model; [26]) and species' propensities to inject venom during a bite (Table 3 for envenoming model; [20,48]). Also, the estimated parameters summarise snake species' biologies and how these relate to human social, economic, occupational and cultural aspects relevant for snakebite epidemiology. For instance, rice paddy farmers are more susceptible to Russel's viper bites (*Daboia russelii*) because rice is usually harvested barefoot in Sri Lanka [49,50]; common krait (*Bungarus caeruleus*) bites occur among the poorest of the poor while victims are asleep on the floor [49]; while *Hypnale spp*. envenoming victims are mostly women who are traditionally in charge of home garden maintenance where this species inhabits leaf litter [51]. These examples show how species' effects on snakebite are intertwined with human socioeconomic and cultural factors.

The second aspect of the decomposed contact rate was the adjustment of snake abundance as a function of land cover and human population density. Here, land cover may represent predominant occupation and socioeconomic status [29], both of which are known to be important snakebite risk factors [22]. Furthermore, we found that the probability that snakebites result in envenoming (Table 3) is also significantly influenced by land cover, especially in urban and agricultural areas. The latter is supported by empirical evidence, as agricultural workers are at greatest risk of snakebite envenoming [22,52]. Consequently, the estimated effects by land cover class summarise snake responses to humans, and effects of human factors such as agricultural occupation and economic status on snakebite and envenoming incidences. All of these characteristics give our model unparalleled forecasting capabilities to close the gap between public health, ecosystem conservation and sustainability.

In contrast with the conceptual and theoretical strengths of our approach, important questions arise to address in future work, for instance: 1) should models be developed with field

data instead of estimates? 2) How does uncertainty and artefacts of incidence estimates affect selection and estimated parameters of our model? 3) Why do highly medically-important species decrease the probability that bites are envenoming according to parameter estimates? For the first question, we infer that a suitable model for field-collected data may be simpler than ours, for which snake abundance estimates in relation to humans should be more robust than currently available [28]. The second point is likely to affect uncertainty of our results, but we interpret the very high similarity between results and estimates as an indication of robustness because most data were independent, apart from human population density, which we discussed above [23]. Finally the negative coefficients estimated for *D. russelii* and *N. naja* mean that, in average, the probability that a bite results in envenoming decreases at increasing abundance of these speceis. However the lack of statistical significance of the estimates (credible intervals contain zero, Table 3), reflect a lack of explanatory power of the spatial heterogeneity, not a lack of biological relevance [53]. Therefore, the contradictory estimates from an epidemiological basis result from the higher envenoming incidence in east than west Sri Lanka (Fig 4) because both *D. russelli* and *N. Naja* are nearly equally abundant on both sides [28]. An alternative ecological explanation for the estimated negative effects for these two very medically-relevant species could arise if the predicted abundances of *N. naja* and *D. russelli* actually serve as proxies for the abundances of non-envenoming species. Abundance surrogacy is relatively common among habitat generalists [54] and non-envenoming snakes are commonly involved in snakebite cases [55], resulting in decreasing risk with the abundance of those non-medically relevant species. In conclusion, given these population-level uncertainties related to the identity of the biting snakes, adequate clinical management in case of a bite should of course be based on expertise and careful clinical evaluation of the envenoming disease.

With careful consideration of the strengths and weaknesses of our approach, we encourage applying and testing our framework in data-poor geographical areas for predicting risk in the absence of other data (e.g., national community survey data) or for testing mitigation interventions. Doing so, however, does require some baseline data as inputs. As mentioned above, the first requirement is a collection of geographical occurrence data of venomous snakes to estimate abundance patterns. Methods and concepts for the analysis of this kind of data in relation to the environment are well established and are described elsewhere (e. g. [56]). Here we used point process models (PPMs) for this purpose given important limitations noted for other common distribution modelling methods (e.g., Maxent) [28,57]. Second is to include some key snake biological/behavioural characteristics for the decomposed contact rates. Relevant traits include aggressiveness, overlap of activity periods with humans [26], venom toxicity to humans and propensity to inject venom after a bite [20]. Lastly, socioeconomic and demographic data may also be used if associated risk factors are well known in the study region. Predictions obtained with the suggested approach will not necessarily represent incidence rates or another measure of burden, but are likely to be broadly correlated with them [28].

Achieving the burden reduction goals laid out in the snakebite roadmap (reducing burden by 50% by 2050; [11]) is an exceptionally ambitious target, requiring advances to our basic understanding of snakebite epidemiology and its treatment. As for other zoonotic diseases, global changes are likely already influencing snakebite and envenoming dynamics, and efficient management will need to accommodate for such changes. However, unlike for many zoonoses, few tools currently exist for snakebite that both shed light on its mechanistic underpinnings and provide avenues for burden mapping and prediction under scenarios of global change or for testing interventions. Studying snakebite as a zoonotic disease has considerably improved our understanding of its epidemiology: it is a dynamic process, its burden is the result of the effects of humans on the abundance of snakes and both affect the burden of snakebite envenoming. The ecological footprint of climate and humans, via land use, represents key

characteristics of local populations that are related to snakebite levels. All these factors, and the nature of snake models used, make our model a simple, yet effective tool to forecast the impacts of global environmental change on snakebite burden. Such exercises are necessary to develop relevant interventions for the present and into the future to solve the snakebite crisis.

## Supporting information

**S1 Fig. Quantile-quantile plots of both non-spatial models.** A) Mass-action model and B) refuge effect model. The curved shape of incidence rates for the refuge effect model indicates that the distribution of incidence was very different from the incidence rates used as data. (TIF)

**S2 Fig. Direct comparison of models.** A) Refuge effect and B) Mass action. (TIF)

**S3 Fig. Geweke convergence diagnostics of the spatial random autoregressive effects of the snakebites model.** Estimates are expected to lie within -2 and 2 for sampling convergence. (TIF)

**S4 Fig. Geweke convergence diagnostic of the spatial random autoregressive effects of the envenoming model.** Estimates are expected to lie within -2 and 2 for sampling convergence. (TIF)

**S1 Table. Upper credible interval of Gelman convergence diagnostic for the spatial mass-action snakebites model.** Upper CI should not exceed 1.05. for sampling convergence. (XLSX)

**S2 Table. Upper credible interval of Gelman convergence diagnostic for the envenoming spatial model.** Upper CI should not exceed 1.05. for sampling convergence. (XLSX)

## Acknowledgments

We thank Dr. Peter J. Diggle and Dr. Christl Donelly for feedback on late stages of the manuscript.

## Author Contributions

**Conceptualization:** Gerardo Martín, Takuya Iwamura, Kris A. Murray.

**Data curation:** Gerardo Martín, Joseph J. Erinjery.

**Formal analysis:** Gerardo Martín.

**Funding acquisition:** H. Janaka de Silva, David G. Lalloo, Takuya Iwamura, Kris A. Murray.

**Investigation:** Gerardo Martín.

**Methodology:** Gerardo Martín.

**Project administration:** Takuya Iwamura, Kris A. Murray.

**Resources:** Takuya Iwamura, Kris A. Murray.

**Software:** Gerardo Martín.

**Supervision:** Takuya Iwamura, Kris A. Murray.

**Validation:** Gerardo Martín.

**Visualization:** Gerardo Martín.

**Writing – original draft:** Gerardo Martín.

**Writing – review & editing:** Joseph J. Erinjery, Dileepa Ediriweera, H. Janaka de Silva, David G. Lalloo, Takuya Iwamura, Kris A. Murray.

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
