## [Decision Letter · Decision Letter 0]

7 Dec 2021

Dear Dr Martin,

Thank you very much for submitting your manuscript "Redefining snakebite envenoming as a zoonosis: disease incidence is driven by snake ecology, socioeconomics and anthropogenic impacts" for consideration at PLOS Neglected Tropical Diseases. As with all papers reviewed by the journal, your manuscript was reviewed by members of the editorial board and by several independent reviewers. In light of the reviews (below this email), we would like to invite the resubmission of a significantly-revised version that takes into account the reviewers' comments, including those provided in the attachment. 

We cannot make any decision about publication until we have seen the revised manuscript and your response to the reviewers' comments. Your revised manuscript is also likely to be sent to reviewers for further evaluation. To facilitate this process, we would like to invite you to ensure that all source data is presented in the manuscript and/or electronic supplements, or referenced therein.

Sincerely,

Ulrich Kuch

Associate Editor

Jean-Philippe Chippaux

Deputy Editor

Reviewer's Responses to Questions

**Key Review Criteria Required for Acceptance?**

**Methods**

-Are the objectives of the study clearly articulated with a clear testable hypothesis stated?

-Is the study design appropriate to address the stated objectives?

-Is the population clearly described and appropriate for the hypothesis being tested?

-Is the sample size sufficient to ensure adequate power to address the hypothesis being tested?

-Were correct statistical analysis used to support conclusions?

-Are there concerns about ethical or regulatory requirements being met?

Reviewer #1: The title of the study "Redefining snakebite envenoming as a zoonosis: disease incidence is driven by snake

ecology, socioeconomics and anthropogenic impacts" is not perfectly matching or at least some sort of misleading to what the method/object/design of this study is.

The manuscript itself deals more with the application of epidemiological models of zoonosis on snakebite data in a certain region, while the title somehow suggests a more fundamental attempt/hypothesis/discussion: whether snakebite is a zoonosis. Regrouping snakebite as a zoonosis would of course broaden the general interest and attention to this devastating neglected disease, what would be highly desirable. Nevertheless, the definition of a zoonosis stated by the WHO is: "A zoonosis is any disease or infection that is naturally transmissible from vertebrate animals to humans" and "A zoonosis is an infectious disease that has jumped from a non-human animal to humans. Zoonotic pathogens may be bacterial, viral or parasitic, or may involve unconventional agents and can spread to humans through direct contact or through food, water or the environment." (https://www.who.int/news-room/fact-sheets/detail/zoonoses)

Snakebite or snake venom neither matches well the medical definition of infection/infectious nor exactly acts as a "transmitted"/"jumping" disease, because it has not been a disease/pathogen to the snake. If snakebite would be suggested as a zoonosis(hypothesis) , this would definitely need more discussion here. Additionally, the sheer applicability of epidemiological models used for zoonosis on snakebites is not a proof of snakebite being a zoonosis itself. These zoonosis models roughly describe the probability of a potentially dangerous vertebrate animal- human encounter, highly differentiated towards socioecological factors etc.. These models can be adapted to lot of non-zoonotic infectious diseases, like (non-vertebrate-animal-) vector borne diseases. If we accept that, like the snake venom, the threat of the animal to the human is not a disease but rather a inherent property/ability of the animal, we can adapt such zoonosis models to a lot of incidents as fatalities by hippopotamus or animal-inflicted car accidents (transmission of physical force/energy), and if we delete the „vertebrate“ we can probably even use a "zoonosis" model for further animal-human encounters as arachnidae/insect stings or lepidopterism.

Reviewer #2: The author used appropriate mathematical formulas to represent zoonotic spillovers for the snakebites in Sri Lanka, which is the hotspot of snakebite envenoming. Several formulas were used to best represent the burden attributed to the association between reservoirs (i.e., snake distribution in their habitat, their nature), their spillover hosts (human density/occupation) and locations. Therefore, the study is appropriately designed to explain the mechanistic eco epidemiological aspects and understanding of snakebite risks relating to demographic data.

To support the hypothesis, the author used previous research to calculate the mathematical formulas for the distribution and abundancy of reservoir hosts (snakes) and spillover hosts (humans). With different methods formulated depending on the condition of snakes and humans, the study is well-articulated to the objectives.

Reviewer #3: Accept

Reviewer #4: Snakebite to be considered a zoonotic disease is a very important study and perspective. The only concern is the data being referred is unclear and more than a decade old. Urbanization, deforestation, increase in population and climate change related untimely season changes have caused environmental changes affecting flora and fauna in many places. Urbanization juxtaposing with farming activities and irresponsible garbage dumping seem to have resulted in a higher number of man-animal conflict cases in countries in Southeast Asia, especially where snake distribution thrives on prey base found abundantly close to human habitations. 

The basis of the study is whether snakebite should be considered a zoonotic disease. Perhaps more weightage should have been given to bites, circumstances, envenomed cases and medical case history of envenomed patients. 

What the authors intend to justify should be succinctly and clearly stated. Many of the statements in the draft are self contradictory and confusing. The analogy regarding low envenoming by D russelii and Naja naja when abundantly found is very different from what is documented regarding both the species in scientific studies and field observations.

**Results**

-Does the analysis presented match the analysis plan?

-Are the results clearly and completely presented?

-Are the figures (Tables, Images) of sufficient quality for clarity?

Reviewer #1: see above/ I do not want to comment

Reviewer #2: The models of magnitude, distribution of snakebites, and envenoming incidence rates were well explained by the author, and this part has matched the analysis plan. The way the results were presented was clear. The author has clearly indicated how the analysis was done and what kind of models were used in the study.

Reviewer #3: Accept

Reviewer #4: What the authors intend to justify should be succinctly and clearly stated. Many of the statements in the draft are self contradictory and confusing. The analogy regarding low envenoming by D russelii and Naja naja when abundantly found is very different from what is documented regarding both the species in scientific studies and field observations.

Given my background, I was not able to appropriately review the formulae for Functional Relationships for human snake contacts.

**Conclusions**

-Are the conclusions supported by the data presented?

-Are the limitations of analysis clearly described?

-Do the authors discuss how these data can be helpful to advance our understanding of the topic under study?

-Is public health relevance addressed?

Reviewer #1: Data-based/Evidence-based medicine is absolutely indispensable for an effective disease control, so providing data, as this study does, is absolutely needed and is especially useful when planning interventions/prevention programs or treating a snakebite without informations regarding the snake. Nevertheless, every snakebite patient still should be assessed/examined/interrogated individually and treated individually. Not every single snake might have read the book when, where an whom to bite, so there might emerge danger when treating cases "prejudiced" by statistics. In my personal opinion, this limit should be mentioned with a few words in the discussion.

Reviewer #2: By describing the formulas, the author explained how mathematical modelling can be fit into epidemiological tools for snakebites. The author considered all the possible factors from both the snake and human sides. The snake sides included different species and their envenoming nature, as well as the aggressiveness of the snake. Human factors include social status, occupation, and culture. With these features incorporated, the conclusion has provided a good sketch of generalised epidemiological models for snakebites.

Reviewer #3: Accept

Reviewer #4: The basis of the study is whether snakebite should be considered a zoonotic disease. Perhaps more weightage should have been given to bites, circumstances, envenomed cases and medical case history of envenomed patients to justify why it should be categorized differently. Instead more effort has gone in snake distribution and contradictory observations of the same species in East and West Sri Lanka.

**Editorial and Data Presentation Modifications?**

Reviewer #1: (No Response)

Reviewer #2: The study is well-written and I would accept it.

Reviewer #3: Minor revisions:

(Line 106 and 107) Inconsistent use of the terms: “snakebite”, “snake bite”.

(Line 108) Consider replacing line with “In future studies would allow us”.

Typos in lines: 302 (“vaulues”) Figure 3 of supplementary material: (“bundances”)

Format issues: Unequal size of text in Table 2. 

Table 3. For clarity, maybe explain abbreviations of “H. spp” - Hypnale spp and “T.trig.” with T. trigonocephalus on the Table 2 description.

Reviewer #4: (No Response)

**Summary and General Comments**

Reviewer #1: see above/ I do not want to comment

Reviewer #2: Many formulas were used in the study, but the author was able to explain them thoroughly. The data presentation and figures are clearly tied to the outcomes

Reviewer #3: GENERAL COMMENTS

• It would be interesting to have more information on how some other ecologic factors had an impact on the model. On lines 324-327, for example the effect of human population on snake abundance is described by land cover classes. Yet it’s important to describe that other species exist in varying degree in such land cover classes and not only humans and that these other species may also have an effect by being predators or prey of the vertebrate host of interest (snakes)

• The authors don’t seem to mention or include in the model the effect of seasonal variability, a well-known factor in snake-human interactions. They seem to address this only somewhat tangentially by considering the alignment of human-snake activity periods. If in fact was part of the analysis, this may warrant some clarification.

• Additional information would be useful to better understand some parameters. Why is “Tea” a land cover class, is it the main crop in the region? Also, it is not very clear how they determined what the most “medically relevant species are” (incidence of accidents, severity of cases, both?)

SPECIFIC COMMENTS / MINOR ISSUES

• (Line 117-118) While it’s quite understandable that they would exclude from the model a frequency-dependent analysis, it could be still applicable as some snakes after biting (not only humans but also prey), can become venom depleted for a few days, which, if following a zoonosis model, might be something to consider.

• (Lines 140-141) The authors mention that the reservoir hosts raster was estimated with data from a 10 to 11-month period. This would be important to clarify as snake activity varies greatly throughout the year and the lack of information of even that month might be important. 

• (Lines 194-195) Statistical Analysis Software should be informed in the methodology according to convention. (Seems to be R and its packages (JAGS) and on line 279 (NIMBLE). Also, there is no information for the type of geographical information software used.

Reviewer #4: The way the manuscript is drafted, it needs major revisions. The authors themselves have expressed the need for further studies to justify a few observations.

The basis of the study is whether snakebite should be considered a zoonotic disease. Perhaps more weightage should have been given to bites, circumstances, envenomed cases and medical case history of envenomed patients to justify why it should be categorized as a zoonotic disease.

PLOS authors have the option to publish the peer review history of their article (what does this mean?). If published, this will include your full peer review and any attached files.

Reviewer #1: No

Reviewer #2: No

Reviewer #3: No

Reviewer #4: No
---

## [Editor Report · Decision Letter 1]

7 Apr 2022

Dear Dr Martin,

We are pleased to inform you that your manuscript 'A mechanistic model of snakebite as a zoonosis: envenoming incidence is driven by snake ecology, socioeconomics and its impacts on snakes' has been provisionally accepted for publication in PLOS Neglected Tropical Diseases.

Best regards,

Ulrich Kuch

Associate Editor

Jean-Philippe Chippaux

Deputy Editor

---

## [Editor Report · Acceptance letter]

3 May 2022

Dear Dr Martin,

We are delighted to inform you that your manuscript, "A mechanistic model of snakebite as a zoonosis: envenoming incidence is driven by snake ecology, socioeconomics and its impacts on snakes," has been formally accepted for publication in PLOS Neglected Tropical Diseases.

Best regards,

Shaden Kamhawi

co-Editor-in-Chief

Paul Brindley

co-Editor-in-Chief
